# Vibration-Based Approach to Measure Rail Stress: Modeling and First Field Test

**DOI:** 10.3390/s22197447

**Published:** 2022-09-30

**Authors:** Matthew Belding, Alireza Enshaeian, Piervincenzo Rizzo

**Affiliations:** 1Department of Electrical and Computer Engineering, University of Pittsburgh, Pittsburgh, PA 15261, USA; 2Laboratory for Nondestructive Evaluation and Structural Health Monitoring Studies, Department of Civil and Environmental Engineering, University of Pittsburgh, 3700 O’Hara Street, 724 Benedum Hall, Pittsburgh, PA 15261, USA

**Keywords:** continuous welded rails, rail neutral temperature, finite element model, machine learning, nondestructive evaluation, structural health monitoring

## Abstract

This paper describes a non-invasive inspection technique for the estimation of longitudinal stress in continuous welded rails (CWR) to infer the rail neutral temperature (RNT), i.e., the temperature at which the net longitudinal force in the rail is zero. The technique is based on the use of finite element method (FEM), vibration measurements, and machine learning (ML). FEM is used to model the relationship between the boundary conditions and the longitudinal stress of any given CWR to the vibration characteristics (mode shapes and frequencies) of the rail. The results of the numerical analysis are used to train a ML algorithm that is then tested using field data obtained by an array of accelerometers polled on the track of interest. In the study presented in this article, the proposed technique was proven in the field during an experimental campaign conducted in Colorado. A commercial FEM software was used to model the rail track as a short rail segment repeated indefinitely and under varying boundary conditions and stress. Three datasets were prepared and fed to ML models developed using hyperparameter search optimization techniques and k-fold cross validation to infer the stress or the RNT. The frequencies of vibration were extracted from the time waveforms obtained from two accelerometers temporarily attached to the rail. The results of the experiments demonstrated that the success of the technique is dependent on the accuracy of the model and the ability to properly identify the modeshapes. The results also proved that the ML was also able to predict successfully the neutral temperature of the tested rail by using only a limited number of experimental data for the training.

## 1. Introduction

With the advent of continuous welded rails (CWR) the problem of extreme compression or extreme tension along rails during hot or cold days became evident. To mitigate such thermal effects, CWR are pretensioned during installation such that the rail neutral temperature (RNT) is within the range of 32 °C to 43 °C. The RNT which is often denoted as *T_N_* is the temperature at which the net longitudinal stress in the rail is zeros. The pretension cannot be higher to avoid fractures during the cold seasons.

Structurally speaking, CWR are often considered as an ideal column, in which buckling may occur when the rail temperature *T_R_* reaches or exceeds the critical temperature *T_cr_*, calculated as:(1)$$\frac{\sigma_{cr}}{E\alpha }+T_{N}=T_{cr}$$

In Equation (1), *α* and *E* are the coefficient of thermal expansion and the Young’s modulus of the steel, respectively, and *σ_cr_* is the Euler stress of the rail steel. Generally, *α*, *E* and *σ_cr_* are known values by design. Therefore, *T_cr_* can be estimated and buckling events can be predicted and prevented once *T_N_* is measured. To this end, *T_N_* can be determined by the longitudinal stress *σ_R_* at any *T_R_* using the equation:(2)$$T_{N}=T_{R}-\frac{\sigma_{R}}{E\alpha }$$

In Equation (2), *σ_R_* is considered positive when the rail is in compression.

Over the years, operational conditions and periodic maintenance decrease the RNT to unknown values, increasing the risk of extreme compression, and therefore thermal buckling. As such, there is a need to measure *σ_R_* and *T_N_*. One of the most common methods consists of instrumenting the rail with strain gage rosette and then cutting the rail to measure the longitudinal strain difference. The rosette is then left in place to measure the pre-tension subsequent to the newly welded segment and the transverse strains occurring hereinafter during normal operation. However, this approach is time-consuming and destructive leading to a long-time quest for in situ cost-effective nondestructive evaluation (NDE) methods to measure axial stress or estimate RNT.

Current NDE methods can be clustered according to the physical principles being exploited, such as electromagnetism [1,2], ultrasounds [3,4,5,6,7], or acoustics [8,9,10,11]. An electromagnetic approach deploys the magnetic Barkhausen noise. This method is premised on the fact that the magnetic permeability in steel material of the rail is positively correlated with the tensile stress. However, the predicted stresses might be adversely affected by the paint coating, unevenness, and residual stress. An accurate lab calibration is also required to apply this technique [1,2].

Nonlinear ultrasonic guided waves transmitted and detected along the web of the rail to of interest were used to calculate a nonlinear parameter that was found to be stress dependent [4,5,6,7]. Another approach based on mechanical waves was developed relying on the correlation between the propagation characteristics of highly nonlinear solitary waves at the point of contact with an axially loaded rail beam [8,9,10,11,12,13]. This technique showed satisfactory results under laboratory conditions [13] but never got the chance to be tested in the field, on real railroads.

NDE methods to estimate the axial stress can also be categorized into static and dynamic approaches [14]. A notable example of former is VERSE^®^, acronym of Vertical Rail Stressing Equipment [15]. This method requires unclipping 30 m of rails from the fasteners. Then, the unanchored rail segment is lifted by a certain amount. The rail deflection under this vertical load is gaged and used as a proxy measure to estimate the axial load in the rail. This longitudinal load is converted into stress using the rail cross section and then plugged in Equation (2) to estimate the neutral temperature [16]. Safety considerations require that VERSE^®^ is used with the rail under tension, i.e., when the temperature of the steel is below the unknown neutral temperature. Another static approach to evaluate the axial stress consists of strain gages that are attached to the web face of the rail, close to the neutral axes of the section [16]. To be effective, the system must remain in place for months if not years. Knopf et al. [17] proposed a method to quantify the axial stress using thermal imaging and a StereoDIC system to obtain different strain components on the web surface of a rail segment. The strain along the neutral axis was then combined with the vertical deformations of the rail caused by thermal loading to estimate the axial load in the rail. This approach was only tested in the lab and verified by some numeric and finite element modeling but, to the authors’ best knowledge, never validated in the field.

Dynamic approaches have attracted numerous researchers. Weaver [18] and Damljanovic and Weaver [19] proposed a method based on the dependency of dynamic vibration characteristics to axial stress. A shaker excited the rail at a specific frequency and the induced displacements were recorded with a laser vibrometer. The rail rigidity and the wavelength of the dynamic response were used to evaluate the stress. The results from the field tests disagreed with the strain gage system present on the tested rail. A dynamic approach based on high-frequency vibrations was proposed by Wu et al. [20]. A steel ball was used as impactor to generate surface waves above 30 kHz, recorded with a polarized condenser microphone. A finite element model (FEM) was generated in support of the experimental efforts. The numerical and the experimental results demonstrated that the waves propagating at 37 and 76 kHz are sensitive to the stress and were used to train a neural network to predict the stress.

Besides the work of Wu et al. [20] there are other examples of machine learning algorithms (MLA) applied in support of rail inspections. Asber [21] proposed an automated visual inspection method to detect faults in different components of a railway. The track was scanned with a camera and the images were fed into a deep neural network, trained to classify different components such as fasteners, ties, etc., and identify missing fasteners. A similar inspection approach based on Convolutional Neural Networks (CNN) was used by Marasco et al. [22] to inspect and classify tunnel defects. In this study the Fourier Transform of the tunnel images were fed into the CNN to identify damages and cracks in the tunnel lining. The input to the CNN was basically profiles obtain by a Ground Penetration Radar (GPR). The results showed around 90% accuracy in damage detection. Enshaeian et al. [23] presented a vibration-based approach to link the axial stress in rail beam with the frequency of vibration relative of a few modes. The method was proven in the lab using a short AREMA 132 rail segment subjected to compression. Aloisio et al. [24] also investigated the vibrations of railway bridges to evaluate the effect of ballast properties on the triggered vibrations. To model the rail and ballast they used beam and spring-damper elements, respectively, and the train effect was simulated by a moving load. That study showed the significant effect of ballast on the load distribution and damping.

In the study presented in this article, the vibration-based approach introduced by Enshaeian et al. [23] was improved, expanded, and validated in the field. An enhanced FE modeling along with certain MLAs were created to model and learn the interdependency of the modal characteristics of CWR to axial stress, rail geometry, and the restraints provided by crossties, ballast, and fasteners. These MLAs consisted of Linear Regression, Extreme Gradient Boosting (XGBoost), CatBoost, Ensembling, Artificial Neural Networks (ANN), Support Vector Machines (SVM), Decision Trees, and Gaussian Process Regression (GPR). It is noted here that the effect of the boundary conditions governed by the supporting components is quite challenging to measure and quantify [18,19]. In addition, construction misalignments such as deviation of tie-to-tie distance compared to the exact design value make the problem even more complex. The numerical model developed in this study provided a comprehensive set of data that frames the interdependency among stress, geometric layout, modal characteristics, and boundary conditions. The synthetic data generated by the FEM were used to train a MLA capable of learning such interdependency. Having trained the model, the axial stress was identified using field data provided in the form of resonant frequency peaks extracted from accelerometer signals. Linking the axial stress in CWR with the modal characteristics of a given rail is not a new idea and has challenged the rail industry for decades because the vibrations are affected by several variables that are not easy to determine empirically. Therefore, the development of an effective and robust MLA gets challenging when it is not trained by reliable data, which in the current study are generated numerically. In addition, finding the optimum set of input features to train the MLA adds to the complexity of the problem.

The supervised learning algorithm developed during a preliminary work [25], was improved and applied to estimate the longitudinal stress of a rail track on concrete crossties. Two input features spaces were considered: one containing the frequencies from the first five lowest modes of vibration; one containing a subset of the former using the two highest modes of vibration. The training set was created numerically using the commercial finite element software ANSYS. The field test involved a 5 degree curved 141 RE rail on concrete ties (Figure 1).

Two tri-axial accelerometers were bonded to the gage side of the rail head, one above a cross-tie and one at the mid-span between two ties. The dynamic response of the rail to the impact of an instrumented hammer along the lateral (sideway) direction was recorded. The impact triggered vibrations along the weak axis of inertia. The experimental frequencies were extracted via Power Spectral Density (PSD). To minimize the effect of noise, the Welch technique [26] was used to calculate the spectral density functions.

The main scientific novelty of this paper is the application of artificial intelligence to a vibration-based approach, including field test validations, for the estimation of longitudinal stress and then RNT. This field test demonstrates the feasibility but also the challenges of the proposed approach to address the long-standing quest for linking the modal characteristics of CWR to the axial stress.

The paper is structured as follows. Next section describes the implemented ANSYS finite element model along with the results of a sensitivity analysis in which some geometric and mechanical parameters of the railroad were varied to quantify their effects on the vibration and dynamic characteristics of the rail. The experimental setup is described in Section 3. Section 4 presents the data preprocessing of the learning algorithms. Section 5 and Section 6 present the ML architectures and the numerical results on the numerical and experimental training data. Lastly, Section 7 ends this article with some concluding remarks and the path of ongoing and future studies.

## 2. Finite Element Model

### 2.1. Implementation

To explain the structural buckling in CWR, some researchers considered any given railroad “equivalent” to a single beam segment of finite length [27,28,29]. Other investigators considered instead two parallel beams of finite length along with linear springs, which represent the lateral resistance associated with the ballast and crossties [30,31,32,33]. In both approaches, the “equivalent” beam(s) are fixed at the ends a comprehensive review of these analyses can be found in Enshaeian and Rizzo [14].

In the present study, the first approach was adopted. A computationally cost-effective finite element approach, based on an irreducible sub-structure model [34,35,36] was implemented in ANSYS. The structurally irreducible component is considered as the smallest track segment that can be replicated to generate miles long rails. This repeating element is also known as the unit cell. The cell included a tie-to-tie long rail at the ends of which there were three translational (*K_x_*, *K_y_*, *K_z_*) and three rotational springs (*K_rx_*, *K_ry_*, *K_rz_*), which mimic the role of fasteners, ties and ballast. The direction of these springs are shown in Figure 2. For example, *K_y_* represents the lateral resistance whereas the coefficients *K_rx_* and *K_rz_* account for the resistance to the torsion along the rail running direction and the lateral bending along the weak axis of the section, respectively.

The rail was modeled using BEAM188 and COMBIN14 elements. The latter are combined spring-damper elements and were used to model the springs. The properties of conventional ductile carbon steel (the same material type used for the tested rail) were used: Young modulus *E* = 206 GPa, Poisson’s ratio *ν* = 0.29, thermal expansion coefficient *α* = 11.5 × 10^−6^/°C and density *ρ* = 7850 kg/m^3^. Considering potential variability in these material properties with respect to standard values above, some sensitivity analysis was performed and the results are presented in Section 2.3. The mesh was chosen to capture accurately the mode shapes associated with all frequencies observed in the field test. The accuracy of the mesh was determined by running a few convergence analyses. It was found that 30 beam elements were sufficient to model the unit cell and no further mesh refinement was necessary.

When the rail steel properties are known, the estimation of the axial stress using vibration-based methods contains 7 unknowns (the six springs and the stress). The coefficients *K_x_*, *K_z_*, and *K_ry_* were fixed because the field test included the observation of lateral-torsional excitations only. They were equal to *K_x_* = 6000 kN/m, *K_z_* = 35,000 kN/m and *K_ry_* = 5000 kN-m/rad, and were taken from Yi et al. [36]. Although they may not be representative of the conditions encountered in the field tests, these three fixed coefficients can only affect the vibrations in the longitudinal direction and deformations along the strong axis of the rail, which are both ignored in this study. The remaining three springs and the longitudinal stress were considered and listed in Table 1. To show the relevance of the ties, rail on wooden ties was also considered and the associated values are lower than those on concrete because timber supports are softer.

For each combination of spring coefficients, thirty-three different longitudinal stresses were considered ranging from +33.6 MPa tension to −33.6 MPa compression, at 2.1 MPa decrements, yielding a total of 14 × 14 × 21 × 33 = 135,828 case scenarios for the concrete track and 6 × 6 × 9 × 33 = 10,692 combinations for the wood track model. Given the selected values of *E* and *α*, the resolution of 2.1 MPa is equivalent to 0.89 °C, and the stress range is comprised between −14.18 °C to +14.18 °C with respect to the neutral temperature.

### 2.2. Results

#### 2.2.1. Model Verification

The finite element model was validated first by comparing its performance with a more conventional approach. Figure 3 illustrates the frequency of the pinned-pinned mode shape of a RE 136 rail beam (RE 136 is a rail section produced according to the 136 lb/yard rail standard set by AREMA, American Railway Engineering and Maintenance of way Association). This figure illustrates the results obtained by both the unit cell approach and the conventional FEM. The pinned-pinned mode is referred to a mode shape of the rail beam in which the nodes are located at the cross ties. A typical 0.6 m tie-to-tie distance was considered. Figure 3a shows the shape relative to mode E (introduced in a later section of this paper) and the corresponding frequency for a 6-m-long rail, whereas Figure 3b compares the frequency found with the two approaches for various rail lengths, and demonstrates that the predicted frequencies converges asymptotically when the real rail is about 30 m long.

The results imply that the use of the unit cell, which includes only a single tie-to-tie segment, provides accurate results without the computational effort required to model 30 m of rail. For more details about the implementation of this structurally irreducible modeling, the interested readers are referred to Urakawa et al. [34] and Yi et al. [36].

#### 2.2.2. Modal Characteristics

The effect of stress on the frequencies of vibration was investigated by implementing a two-steps static-eigenvalue analysis. The first step consisted of pre-stressing the rail by exerting a concentrated force along the longitudinal direction and adapting the stiffness matrix accordingly. Subsequently, the eigen value (modal) analysis was carried out. Some results are presented with Figure 4.

The left column shows the mode-shape of the first five lowest modes, labeled as B, D, E, F, and G, of a 0.610 m long rail. This length corresponds to the center-to-center crosstie measured in the field. The mode-shape notation follows Wang et al. [35] and Yi et al. [36]. Mode B is mainly translational while mode E is flexural-torsional. Bending dominates mode E and mode F, whereas modes B and D are basically rigid modes of the rail below 400 Hz, and are less affected by the axial force. Mode E and mode I (not shown here) are the 1st order lateral and vertical pinned-pinned modes, respectively. Mode G is similar to mode E, while the bending deformation of mode G is significant at the rail bottom, and the rail head and bottom are subject to antiphase motion. The mode shapes presented in Figure 4 were obtained by assuming *K_y_*, *K_rx_*, and *K_rz_* equal to 7000 kN/m, 116 kN-m/rad and 5600 kN-m/rad, respectively. The values of *K_y_*, *K_rx_*, and *K_rz_* were selected because the corresponding mode frequencies are close to the experimental values found in the field.

Fixing *E*, *α*, and the spring coefficients along all directions, the individual effect of the stress (and therefore temperature) on the frequencies of the five modes was evaluated. The results are presented in the second column of Figure 4. Negative *σ*, i.e., positive ΔT, indicates compression, whereas positive *σ*, i.e., negative ΔT, indicates tension. For example, the graph on the first row/second column shows the frequency of vibration as a function of the stress and of the corresponding temperature increase/decrease with respect to the RNT at certain values of the six stiffnesses. The regression of the numerical results shows that mode B is barely dependent on the stress, and its natural frequency of vibration is expected to change only 0.002 Hz per degree Celsius. The same trend, i.e., stress-independence, is seen for mode D. On the contrary, modes E and F are the most sensitive modes to the temperature change. The vibration of mode E is expected to change by 1 Hz every 5 degrees Celsius, equivalent to 0.084 Hz per MPa change in stress.

Notably, these slopes are nearly identical to those reported by Urakawa et al. [34] and Yi et al. [36]. However, with respect to those works, this study expands the state-of-the-art knowledge by investigating the impact of both stress and boundary conditions. Previous studies were mostly focused on the stress influence whereas here the impact of boundary conditions on the resonant frequencies is also included.

The effect of the boundary conditions, namely *K_y_*, *K_rx_*, and *K_rz_*, at the neutral temperature is shown in the rightmost column of Figure 4. The charts demonstrate that modes B, D, and F are affected by coefficients *K_y_* and *K_rx_*, whereas modes E and G depend on *K_rz_* only. For example, for high values of stiffness *K_y_*, the natural frequency of mode B is strongly dependent on *K_rx_*, such that the frequency can range from 100 Hz to 180 Hz. On the contrary for lower values of *K_y_*, the frequency of vibration of mode B remains nearly constant and equal to 62 Hz or 68 Hz, regardless of *K_rx_*.

To provide a broader view about the effect of geometry and boundary conditions, Figure 5 illustrates the same analysis applied to a 136 RE track on wooden ties. As said earlier, softer spring coefficients were considered. In addition, the tie-to-tie distance was reduced to 0.540 m consistent with typical field test conditions. The results are qualitatively similar to those relative to the concrete ties. Modes B and D are stress independent, whereas modes E and F are the most sensitive ones to the change in rail temperature. Mode F is only slightly affected by *K_y_* variations, as different curves associated with *K_y_* coefficients almost overlap to each other.

### 2.3. Sensitivity Analyses

To quantify the effect of geometric or mechanical parameters such as the center-to-center crosstie distance, the geometry of the rail cross-section, and the steel properties on the rail vibrations, the following sensitivity analyses were conducted numerically under the assumption that the rail was stress-free.

Figure 6 shows the results relative to two center-to-center distances, namely 0.610 m and 0.540 m, for a 141 RE rail section and ranges of coefficients *K_x_*, *K_z_* and *K_ry_* used in Figure 4. The figure shows that the distance between two consecutive cross-ties has almost no effect on modes B and some minor influence on mode D, but causes a significant increase in the frequency of modes E, F, and G. Modes B and D are mainly rigid body motions with negligible internal deformations. The slight increase in frequencies is attributed to the higher distributed resistance of the support springs per unit length. Instead, the energy needed to trigger the flexural modes, such as E, is a function on the flexural energy of the beam itself rather than the spring coefficients at both ends. The corresponding mode shapes experience higher curvatures compared to the lower modes. As higher curvature corresponds to higher flexural strains and stresses, these modes require more energy to be triggered. Therefore, much higher external energy is required to excite these higher modes as the length reduces, which in turn leads to a significant increase in frequency.

The effect of the rail cross section was investigated by comparing the frequencies of vibration of a 0.610 mm long 136 RE rail and 141 RE rail at given boundary conditions. The results are summarized in Table 2. The frequency of mode B and mode D decreases for the larger rail.

The opposite trend is seen for the flexural modes E and F. The reason behind this apparent conflicting result is also associated with the mode-shapes. The stiffness of modes B and D do not contribute to the overall stiffness relative to the deformed mode shape. As a result, their frequency can be characterized by the spring coefficients at both ends and the mass of the rail beam. By increasing the rail section from 136 to 141 RE, the mass rises by almost 3%. Besides, since the frequency in inversely proportional to the mass (ω = √(k/m)), this higher amount of mass causes the observed reduction in the frequencies of modes B and D. On the other hand, the mainly flexural modes E and F are subjected to both mass increase but also stiffness against flexure increase. The area of the section and the moments of inertia have length^2^ and length^4^ dimensions, respectively. Since the inertia and consequently, the flexural stiffness, is a higher order function of the section dimensions compared to the area and mass, the rise in the stiffness overshadows the mass increase, which results in the ascending frequencies in the third and fourth row of Table 2.

Finally, the combined effect of the Young modulus, Poisson ratio, and density of the steel was examined by selecting two values for each parameter, namely 206 GPa and 210 GPa, and 0.27 and 0.29, and 7850 and 7650 kg/m^3^, respectively. The eight cases were applied to a 141 RE rail under certain fixed spring coefficients, and the results are presented in Figure 7. To facilitate the interpretation of the results, the vertical range of the plots was left equal to 16 Hz.

Figure 7a,b show that modes B and D are not affected by the modulus and the Poisson’s ratio, whereas they are slightly affected by the density. These results are consistent with what shown in Figure 6a,b. The effect of density is also consistent with what seen with the cross section; lowest density yields to highest frequency. Figure 7c,d show the obvious dependency of flexural modes E and F on the Young modulus as well as the Poisson’s ratio. Higher modulus causes higher frequencies, whereas lower Poisson’s ratios have the same effect. Analogous to subplots (a) and (b), reduction in the density and mass of the rail beam, results in higher frequencies.

## 3. Field Test

### Setup

The field test was conducted at Transportation Technology Center (TTC), in Pueblo, Colorado on a 5° curved RE 141 rail on concrete ties showed in Figure 1. A close-up view is presented in Figure 8. Two tri-axial accelerometers were bonded to the railhead above a tie and at midspan. Both accelerometers were triggered via an instrumental hammer and sampled at 10 kHz. Two thermocouples were attached to the rail, one on the head and one on the web. The rail location was also equipped with strain gauges and temperature measurement devices operated independently by TTC.

Figure 9 shows the temperatures recorded by the three independent systems during the first day of test (day 1). The recordings from TTCI were taken automatically at regular intervals, whereas the recordings from the Pitt system were taken manually every time the rail was hit with the hammer. Impacts were applied on the field side of the rail just behind the midspan and tie location where the sensors were bonded. Only one of the two spots was hit during a single measurement. The trend is identical for all three systems. The readings from the thermocouple attached to the rail head are biased by solar irradiation and scattered by the presence of clouds. The readings from the web are lower than those on the head, and those recorded by the TTC system were about 1–2 °C higher than those recorded by Pitt thermocouple. All systems recorded an increase of temperature in the order of 15 °C between 10 AM to 12:30 PM after which the increase was milder. The rail started cooling after 2 PM.

For illustrative purposes Figure 10a shows a time series associated with the lateral vibration measured by the accelerometer located at the cross-tie. The corresponding power spectral density is presented in Figure 10b along with the spectrum of the vertical acceleration. After zero padding, the resolution of the PSD was 0.3 Hz. Several peaks are visible around 150 Hz, 220 Hz, 350 Hz, and 550 Hz.

Two additional frequencies, around 490 Hz and 500 Hz, that not observed at the cross-tie were seen at the midspan. These six frequencies were consistent across both days of testing and were selected and labelled as F1, …, F6 to be the possible features for the ML models discussed later in this article. For peaks that were not as obvious or clear and therefore were not able to be extracted, they were locally linearly interpolated using the surrounding two to three immediate hits at that frequency. These peaks are placed for reference in Table 3.

For illustrative purposes, Figure 11 shows the experimental frequencies found on day 1 relative to the peak around 500 Hz for all the impacts before 12:30 PM with respect to the rail web temperature recorded by the Pitt system.

The plots results confirm the numerical prediction, i.e., the inverse correlation between the frequencies and rail temperature. The slope of the linear regression was equal to −0.17 and −0.43 Hz/°C for the peaks around 500 and 530 Hz, respectively. Notably, the number of data points in Figure 11 is lower than the number of data points in Figure 10a because some hammer hits did not trigger the 500 Hz mode.

## 4. Machine Learning: Preprocessing

### 4.1. Datasets

Three datasets, summarized in Table 4, were created in support of the ML.

The first dataset, used strictly for training/validation, included a subset of the ANSYS database with the same variables, i.e., spring stiffness and longitudinal stress. It included frequencies associated with modes B, D, E and F as predictors and an output of stress. Mode G was ignored because it is difficult to trigger with a hammer. The ML was designed to learn the interdependency between the modal characteristics and the boundary conditions of the rail, and the effects of the temperature. Concisely, all discussed and presented graphs relative to the support conditions (Figure 4, Figure 5, Figure 6 and Figure 7) were deployed implicitly in the ML for the first dataset. By removing half of the intermediate *K_y_* and *K_rx_* conditions, the variability in the experimental frequencies, discussed in a later section, could be properly compensated against their respective boundary conditions. The result of this discretization reduced the dataset from 135,828 data points to 33,957. Hereinafter, this dataset is referred to as RFEA (reduced finite element analysis) and was used to predict the axial stress and then infer the RNT with Equation (2).

Six experimental peaks were extracted from the PSD of each time-waveform recorded by the two accelerometers. These peaks and the corresponding temperature readings formed the second and third datasets. The main difference between the numerical and the experimental datasets is that for the latter, the output is a ratio *β* of the RNT to the rail temperature, i.e., *T_N_* = *β T_R_*.

For Day 1, 40 hammer impacts were used for training/validation. The frequencies from Day 2 were used as the test case. Hereinafter, the experimental dataset is referred to as D1 whereas its corresponding testing set is referred to as D2 (Day 2).

The third dataset clustered together the experimental data from both days, making a total of 93 data points to be split for training/validation/test. The same inputs and term *β* was used for the output. This dataset is called D12 (Days 1 and 2).

### 4.2. Mode Shape Classification

Linking any experimental frequency peak to the correct mode-shape was challenged by the limited number of accelerometers available in the field. Thus, shape labelling was necessary to assign the empirical peaks to the appropriate input variable for the numerical model. To address this challenge, the experimental frequencies were sifted through the numerical prediction to gauge the probable mode shape. Figure 12 shows the 135,828 numerical values of the frequency of vibration relative to modes B, D, E, and F overlapped by the six empirical frequencies found in the frequency domain of the accelerometers data. The figure demonstrates that modes E and F are mostly overlapping and span over a relatively small range of frequencies. Conversely, modes B and D are mostly separated with mode D spanning from 100 to over 450 Hz. The figure also shows that the experimental frequency around ~220 Hz does not belong to almost any specific mode shape.

To emphasize the challenge, Figure 13 shows the frequencies of modes B, D, E, and F against themselves. The overlap between mode E and F, as well as with parts of mode D, is visible.

### 4.3. ML Regression Model Generation

The prediction of the longitudinal stress and of the ratio *β* constitute supervised learning tasks. Several regression machine learning methods were considered using the Python *MLJAR* library and Matlab’s machine learning toolbox. The methods were Extreme Gradient Boosting (XGBoost), CatBoost, Ensemble, Artificial Neural Networks (ANN), Linear Regression, Support Vector Machines (SVM), Decision Trees, and Gaussian Process Regression (GPR). Results of each model were compared amongst each other, and the best was chosen. For the sake of completeness, some of these models are briefly summarized here.

Linear Regression is a traditional approach with low flexibility and easy interpretation when compared to other methods. It also has an advantage of being much easier to train compared to some higher flexibility models but can come at a cost of being highly biased if the function to be mapped is nonlinear. Linear regression is expressed as:(3)$$Y=\beta_{0}+\beta_{1}X_{1}+\ldots +\beta_{n}X_{n}+\varepsilon$$
where *Y* is the response, *X_i_* is the i-th predictor, *β* is an association variable between the response and the i-th predictor, and *ε* is an error term.

GPRs extend upon Equation (3) by taking on a non-parametric probabilistic approach to regression. With Equation (3) in a simplified form, *Y* = *f*(*X*) + *ε*, the Gaussian regression models the function *f*(*X*) as a Gaussian process. The function *f*(*X*) is jointly distributed with a mean function, *m*(*X*), as well as a covariance function, *k* (*X*, *X*’). Each of these functions is defined below:(4)$$m\left(X\right)=E\left[f\left(X\right)\right]$$
(5)$$k\left(X,X^{\prime }\right)=E\left[\left(f\left(X\right)-m\left(X\right)\right)\left(f\left(X^{\prime }\right)-m\left(X^{\prime }\right)\right)\right]$$

These make up the Gaussian process described by *f*(*X*)~GP(*m*(*X*), *k* (*X*, *X’*)) where *m*(*X*) is often zero. A variety of covariance functions or kernels as they are often referenced as are available to try. This article explores the *squared exponential*, *Matern 5/2*, *exponential*, and *rational quadratic kernels*, which are defined in Equations (6)–(9).
(6)$$k\left(X_{i},\;X_{j}\right)=\sigma_{f}^{2}exp\left\{\frac{-1}{2}\frac{{\left(X_{i}-X_{j}\right)}^{T}\left(X_{i}-X_{j}\right)}{\sigma_{l}^{2}}\right\}$$
(7)$$k\left(X_{i},\;X_{j}\right)=\sigma_{f}^{2}\left(1+\frac{\sqrt[{2}]{5}r}{\sigma_{l}}+\frac{5r^{2}}{3\sigma_{l}^{2}}\right)exp\left\{\frac{-\sqrt[{2}]{5}r}{\sigma_{l}}\right\}$$
(8)$$k\left(X_{i},\;X_{j}\right)=\sigma_{f}^{2}exp\left\{\frac{-r}{\sigma_{l}}\right\}\;$$
(9)$$k\left(X_{i},\;X_{j}\right)=\sigma_{f}^{2}{\left(1+\frac{r^{2}}{2\alpha \sigma_{l}^{2}}\right)}^{-c}$$

Here, *σ_f_* is the signal standard deviation, *σ_l_* is a characteristic length parameter for how far *X*’s can be from their target *Y* to become uncorrelated, and *c* is a scale-parameter as well that can only be positive. Although four common basis functions are available to use, the constant basis is only utilized in this study. Marginal log likelihood is maximized in order to estimate the parameters of the GPR.

Decision trees can be used for both classification and regression tasks. Trees act like yes/no questions and are also very easily interpretable compared to other methods such as neural networks. The basic breakdown of a decision tree for regression can be summarized by splitting the predictor space into N non-overlapping regions and making the same prediction for any observation that falls into that region. The prediction is given by the mean of the training observations provided in the split region. One drawback of this method is the tendency to easily overfit, which can be alleviated via pruning methods [37].

ANNs take on some of the most complex tasks thanks to their extreme flexibility. Such tasks include object tracking and recognition, natural language processing, and regression [38,39,40,41,42]. They are loosely based on the human brain and learn via the backpropagation algorithm, allowing for each individual neuron’s error to propagate through the structure. We utilize a neural network architecture for our mode shape classifier.

SVMs are often used for classification tasks but can be applied to regression as well. They attempt to find the ideal hyperplane in an N-dimensional space of features that maximizes the distance between observations. This is based on the maximal marginal classifier which works in cases where the separating plane exists. SVMs have been useful in areas such as damage assessment on harbor caissons [43] as well as health-related regression tasks for predicting the severity of Parkinson’s disease dementia [44].

Ensemble learning strives to make better predictions from a diverse set of models. There are several simple as well as more advanced techniques that exist to accomplish this. From the less complicated side, averaging and max voting can be used on the outputs of several trained models to infer an overall prediction. These two techniques do not produce a new model whereas the more advanced techniques of ensemble learning such as stacking and boosting do. Stacking utilizes the predictions of multiple models as it’s features to create a new model used to predict. Boosting on the other hand, is a sequential technique that uses every succeeding model to correct the errors from the previous model. Once satisfied with the number of correcting models, a weighted mean of all models is taken to develop a strong learner.

### 4.4. Model Evaluation Metrics

With the aim of predicting the field test data via the finite element model, the RFEA was trained via 5-fold cross validation and tested on Days 1 and 2. D1 and D12 also used 5-fold cross validation with D1 testing on D2, and D12 on a 75/25 train/test split. Five random shuffled splits were generated and the respective results of those splits was averaged to obtain the best overall model for D12. K-Fold cross validation was applied because it has the advantage of effectively evaluating predictive models by dividing up the dataset into k partitions and producing different “folds” of training and validation data each time. This ensures every observation has a chance of appearing in the training and validation sets.

Four statistics were calculated to determine the best model: Mean Squared Error (*MSE*), Mean Absolute Error (*MAE*), Root Mean Square Error (*RMSE*), and *R*^2^. Mean Absolute Error (*MAE*), Mean Squared Error (*MSE*), *R*^2^, and Root Mean Square Error (*RMSE*). They are defined as:(10)$$MSE=\frac{1}{n}\sum_{i=1}^{n}{\left(Y_{i}-\hat{Y_{i}}\right)}^{2}$$
(11)$$MAE=\frac{1}{n}\overset{n}{\underset{i=1}{\sum }}\left|Y_{i}-\hat{Y_{i}}\right|$$
(12)$$RMSE=\sqrt[{2}]{\frac{\sum_{i=1}^{n}\left(Y_{i}-{\hat{Y}}_{i}\right)}{n}}\;$$
(13)$$R^{2}=1-\frac{RSS}{TSS}$$

Here, *n* represents the number of samples, *Y_i_* is the target, and *Ŷ_i_* is the predicted value of the model. In Equation (13), RSS represents the sum of squared residuals and TSS the total sum of squares. *MAE* was considered because it measures the mean absolute residual error between the predicted value and target. This does not penalize outliers like *MSE* does, which calculates the mean square residual error between the predicted value and target. R^2^ represents how close the data is to the fitted line acting as a measure of the variation that is covered by the model. *RMSE* is another measure of residual error for evaluating alongside *MAE*, *MSE*, and *R*^2^ to produce a well-trained model. All of these metrics are best minimized with the exception of *R*^2^ which a score of 1 or −1 is indicative of a perfect correlation. The mean absolute error in degree Celsius is also provided to give a clear understanding of the error with respect to temperature.

## 5. Machine Learning: Training Results

Model performance from training is an essential aspect to gauging the ability of the model to predict new unseen data. As mentioned earlier, three datasets were used to estimate the RNT. Two datasets used experimental data collected in the field for training while the other is synthetically driven by modelling.

### 5.1. Mode Shape Classifier

The mode shape classifier was built using a custom neural network designed for one input (frequency) using the existing RFEA dataset. The output is the probability of said frequency being one of four classes: mode B, D, E, and F. This architecture, seen in Figure 14a, contained three hidden fully connected layers with 150, 100, and 50 nodes, respectively, as well as batch normalization for regularization and *ReLu* activations after each hidden layer. The last layer contained four nodes for each class and used the *softmax* function
(14)$$\sigma {\left(\vec{z}\right)}_{i}=\frac{e^{z_{i}}}{\sum_{j=1}^{N}e^{z_{j}}},$$
to generate a probability.

The ANN was 83.1% accurate with its most misclassifications between modes E and F, as seen in the confusion matrix of Figure 14b. This is expected given the overlap discussed in Figure 12 and Figure 13. Nevertheless, this represents an easier methodology to estimating the frequency mode shapes as opposed to sifting through the ANSYS data given limitations in the number of sensors available.

### 5.2. RFEA Model

The python machine learning library *Mljar* [45] was used to train numerous models using four modes as input: B, D, E and F. *Mljar* is an automated machine learning framework for tabular data capable of algorithm selection, feature engineering, model training/optimization, explanation, and evaluation. Given the extensive features it provides in its framework, models were able to be exhaustively trained and analyzed to achieve a well-fit model with respect to the numerous conditions present in the RFEA dataset. Algorithms used were Linear Regression, Random Forest, LightGBM, XGBoost, CatBoost, ANN, and Ensembling. Random search is utilized to optimize their hyperparameters. As said earlier, modes B and D are stress-independent but contribute information regarding the boundary conditions which are necessary to narrow down the search region for stress prediction. As the RFEA dataset contains several boundary conditions, the model accounts for those conditions in the prediction.

Modes E and F are strongly dependent on the axial force. Forty-three models in total were trained through the training process using *Mljar*’s golden features option for feature generation and five-fold cross validation. The best model is shown in Figure 15 alongside its metrics in Table 5. The model is an ensemble that contains five gradient boosted tree models using CatBoost and LightGBM found via the greedy search algorithm. A simple weighted average is used to aggregate their predictions.

### 5.3. D1 Model

The set of data from the first day of testing contained the frequency peaks from 40 hammer impacts. These frequencies varied minimally under constant temperature and can thus be utilized to train with single points at varying rail temperatures. Two sets of frequency inputs were created: F1 through F6, and F5 with F6. These were chosen to investigate the impact of using all available significant features (peaks) as well as using the peaks that are most linked with stress (higher ones). In all, 48 models were tested through five-fold cross validation and evaluated using the four metrics discussed in Section 4.4. The best model for each of these cases is summarized in Table 6. Aside from Matlab’s numerical optimization for training, hyperparameters were not optimized in these models. Parameter exploration went as far as the four GPR kernels discussed in Section 4.3 and width/depth for ANN.

Both the Gaussian Process Regression and SVM were extremely accurate at predicting the scaling factor *β*. To estimate the error, a *β* of 1 (zero stress conditions) was used at a rail temperature of 30 °C. Both produced errors under 0.3 °C for the validation sets. Notably, the model with only two frequency inputs (F5 and F6) outperforms the results of the model with six frequency inputs. This is attributed to noise introduced by the other frequencies as they contain significantly more variance than F5 and F6, and to the fact that F1 and F2 are stress-independent. The predicted validation data can be seen mapped against the true values for both models in Figure 16.

### 5.4. D12 Model

This model focuses on catching any potential data drift from Day 1 to Day 2 to frequency peaks. Although frequencies were nearly identical from Days 1 to 2, there was a minor number of points that were unseen on Day 1 compared to Day 2. An example of this can be seen when comparing frequency 6 (~550Hz) in Day 1 and Day 2 in Figure 17. As shown, frequencies below 545.5 Hz were not observed in Day 2 data whereas Day 1 has no data above 550 Hz.

As done for the data from the first day of test, the same two separate combinations of inputs were trained. Optimization and parameter exploration also remained the same. The results of the best models are summarized in Table 7.

The two-frequency combination of F5 and F6 provided the best results with respect to error and MAE to predict the scaling factor *β*. There is a 28.23% and 40.94% increase in error for each model, respectively, when considering the whole set of data from Days 1 and 2. This is expected as the model maps a larger distribution of data with respect to D1. Note that the first model using all six frequencies still uses a GPR Model whereas a GPR achieved the best generalized performance for the F5/F6 combination. Both models achieve nearly identical performance overall demonstrating a need for only two frequencies. The Validation Predicted vs. Actual plot for these two models can be seen in Figure 18.

## 6. Machine Learning: Testing Results

The model with the RFEA dataset relied on the correct mode shape labelling to predict on empirical frequencies found in D12. In order to realize this, mode shape classification is done.

Figure 19 shows the probability to associate a given frequency, and therefore a given label, to a given mode. The plots show that F3, F4, and F6 (350 Hz, 490 Hz, 550 Hz) can confidently be classified to Modes D and F. Uncertainty, however, does exist with F1, F2, and F5 (150 Hz, 220 Hz, 500 Hz) with probabilities residing between 40–60%. This is not surprising given the overlap demonstrated between modes in Figure 13. Due to the confidence in modes D and F, F3 (350 Hz) is classified as Mode D and F6 (550 Hz) as Mode F. Given these classifications into consideration, Mode B and E are classified as F2 and F5 (220 Hz and 500 Hz), respectively. Note that the B and E classifications can be made due to natural frequencies belonging to one unique type of mode in the RFEA dataset.

### 6.1. Numerical Dataset

The ANSYS data were used to predict the neutral temperature based on the labelling of the experimental frequencies. The results are presented in Figure 20 where the ML predicted temperatures are overlapped to the estimation of the neutral temperature provided independently by TTC. A total of 93 test points are presented. The predictions are based on the following associations: F2→mode B, F3→mode D, F5→Mode E, F6→mode F.

Several considerations can be made from Figure 20. First, the figure shows that the algorithm performs well when the rail was in compression but is heavily biased by the temperature of the rail otherwise. The bias originates from a difference in distributions between the RFEA training dataset and the empirical test set, D12. Therefore, for some data points, the model predicts nearly zero stress values. As a result, the application of Equation (2) which is correct only under the assumption of the column-beam theory introduces the observed bias in Figure 20. Nevertheless, this model achieves a MAE of 2.42 °C and 1.91 °C for day 1 and day 2, respectively, demonstrating promise in utilizing an FEA-trained ML method to predict RNT from vibrational frequencies. Notably, the RNT measured in day 1 is a few degrees Celsius higher than in day 2, consistent with the fact that the rail was overall warmer during the first day of testing. Finally, Figure 20 confirms that the RNT is a function of the rail temperature, which has daily fluctuations.

### 6.2. D1 Results

The numerical database describes a wide range of field scenarios including boundary conditions. However, there are two main limitations. First, the number of case scenarios may be too large to allow an accurate estimation of the correct target when there is a limited number of available input features. The second drawback is that the dataset does not contain any real data and therefore it may still be an approximate account of the real rail.

Leveraging upon the above considerations, the same MLA presented earlier was trained and tested with only experimental data. In this section, the use of values from day 1 to predict the neutral temperature on day 2 is presented. The performance metrics of D1 for both of its respective models is summarized in Table 8. D1 shows a significant ability to generalize to Day 2 and does so at an average error of only 0.89 °C and 0.55 °C. It also demonstrated much less variability in its predictions compared to the RFEA when provided slightly varying frequency inputs. This is evident in Figure 21, where the predicted RNT is plotted alongside the actual RNT.

### 6.3. D12 Results

Section 6.2 showed that the experimental values to train the ML were effective at predicting the RNT. However, D1 still suffers from not being completely representative of the data seen on Day 2. Although most frequencies are accurately covered by the training data on D1, it can be further improved by training on a more complete sample drawn partially from Day 2 as well. This is verified in Table 9 where the average error using five frequencies and two frequencies goes as low as 0.21 °C and 0.16 °C.

This is a nearly 4.2-fold and 3.4-fold improvement in error compared to D1, and a ~12-fold improvement over RFEA for Day 2. Such a significant improvement over the RFEA results can be attributed to a better representation of the actual rail conditions by empirical data compared to finite element simulation. In other words, the distributions of the experimental data and simulated data do not align perfectly. Figure 22 compares the predicted neutral temperature to the estimated neutral temperature provided by TTCI. Because the values of the RNT differ between day 1 and day 2, this plot was preferred over the graphs like Figure 21.

## 7. Conclusions

This article expands upon the work presented at the recent European Workshop on Structural Health Monitoring [46] and proposes a vibration-based approach supported by artificial intelligence and finite element modeling to provide non-invasive reliable estimates of the neutral temperature of continuous welded rails. This approach consists of monitoring rail vibrations and the extraction of the associated modal characteristics to evaluate the axial stress using a machine learning algorithm trained with synthetic finite element data. The feasibility of this technique was proven in the field at a testing facility in Colorado.

An additional strategy was also developed for cases in which numerical data are not available while benchmark experimental data collected from the field become available.

The results demonstrated that both strategies with numerical data or experimental benchmark data are quite promising especially the one based on field data only. An error of 0.54 °C and 0.10 °C with respect to the true neutral temperature measured by an independent party was achieved using extremely limited data and two frequencies inputs compared to the finite element case. The results proved that the data collected during a day worth of testing, were sufficient to estimate the neutral temperature during the second day. The strategy based on finite element data showed promise in its independence from real data. However, it has been also shown that such approach may be negatively biased by the applications of the equations related with the ideal beam-column theory (see Equation (2)) and by the approximation in simplification of the boundary conditions.

Future investigations shall take two directions. The first direction shall improve the model of the rail by replacing the six single springs at the end of the rail cell with more distributed cells. The second direction shall consist of collecting more field data to validate the accurate estimations found in this article by feeding the artificial intelligence with field data only.

## Figures and Tables

**Figure 1 sensors-22-07447-f001:**
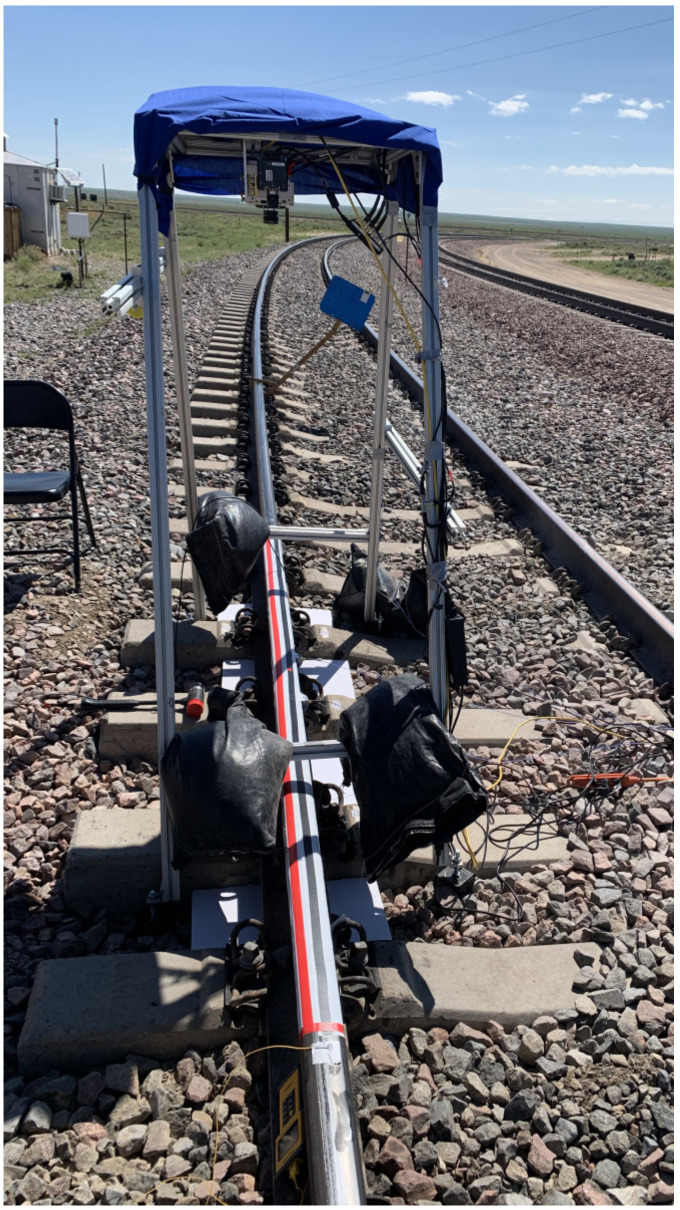
Photo of the railroad section tested in Colorado.

**Figure 2 sensors-22-07447-f002:**
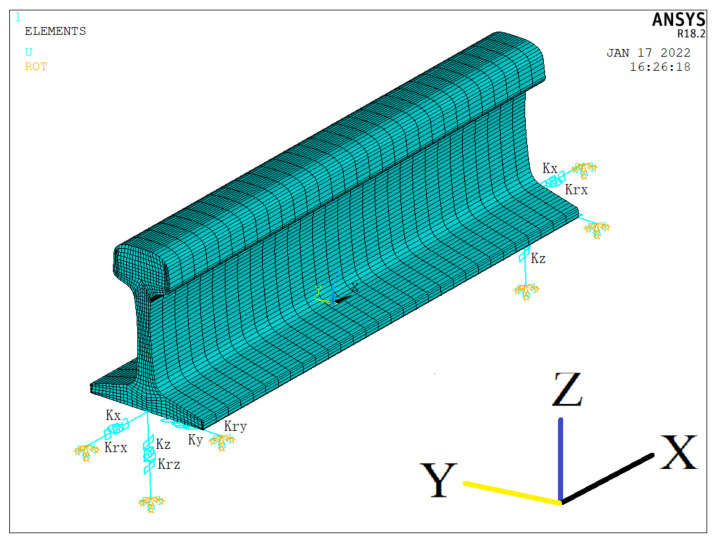
Schematics of the FEM unit cell implemented in this study along with the springs used to model the effects of the fasteners, crossties, and ballast.

**Figure 3 sensors-22-07447-f003:**
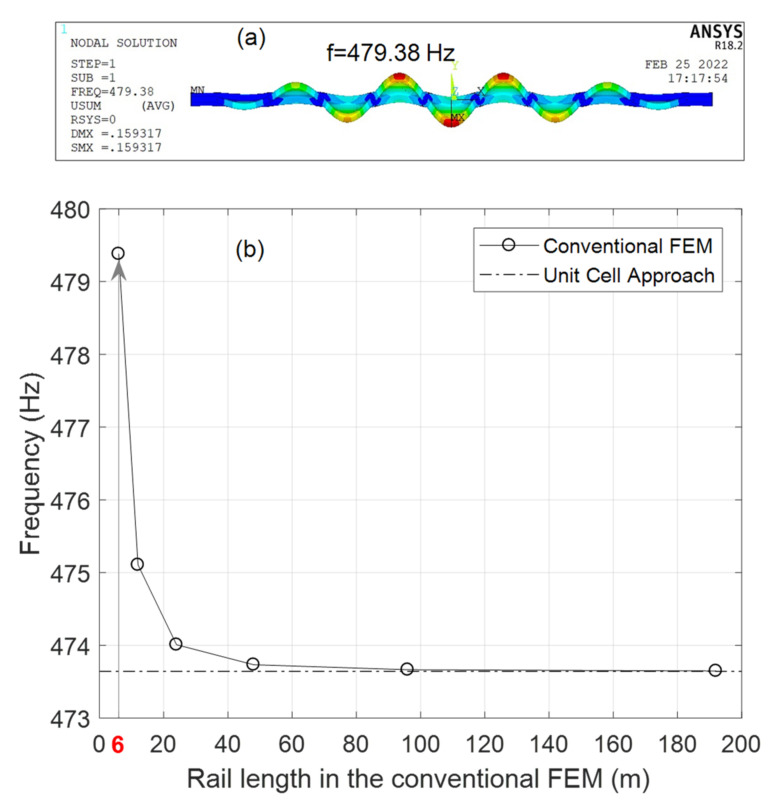
Comparative analysis of the performance of the unit-cell formulation to a conventional finite element approach. The analysis involved a 6-m long rail under pin-pin condition. (**a**) Mode-shape and corresponding frequency obtained with a conventional approach. (**b**) Frequency vs. length of the simulated rail using the conventional FE model and the unit-cell approach.

**Figure 4 sensors-22-07447-f004:**
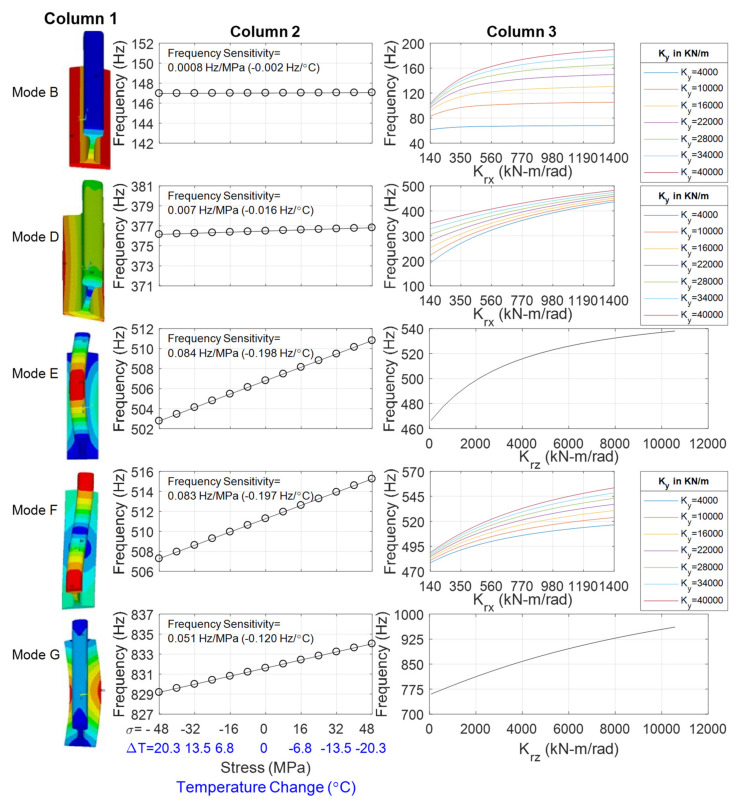
Finite element modeling of a RE 141 tangent track on concrete ties. **Left column:** mode shape of the lowest five modes. **Center column:** frequency as a function of stress and temperature change (with respect to the RNT) at given boundary condition. **Right column:** frequency as a function of certain boundary conditions at neutral temperature.

**Figure 5 sensors-22-07447-f005:**
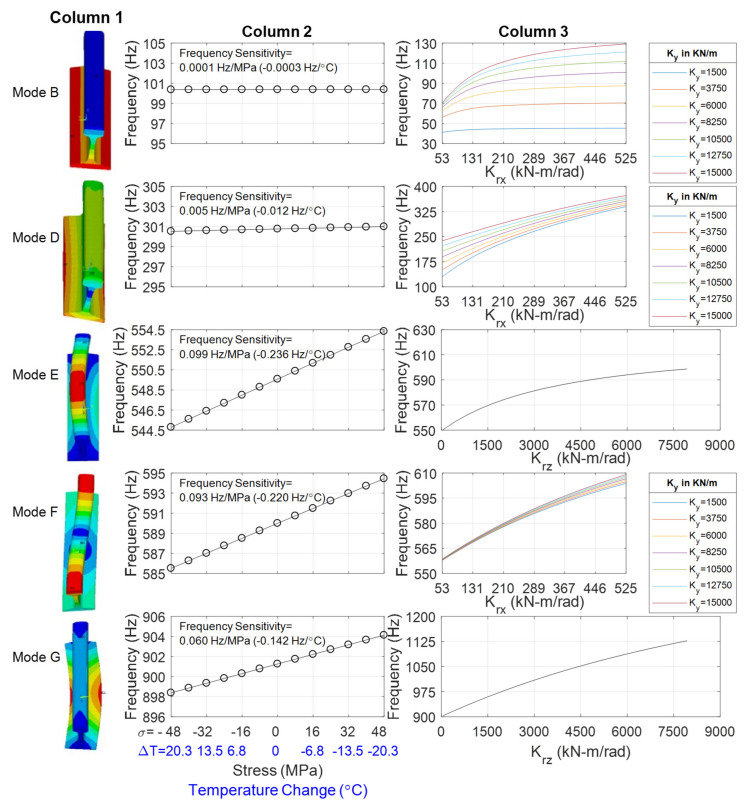
Finite element modeling of a RE 136 tangent track on wood ties. **Left column:** mode shape of the lowest five modes. **Center column:** frequency as a function of stress and temperature change (with respect to the RNT) at given boundary condition. **Right column:** frequency as a function of certain boundary conditions at neutral temperature.

**Figure 6 sensors-22-07447-f006:**
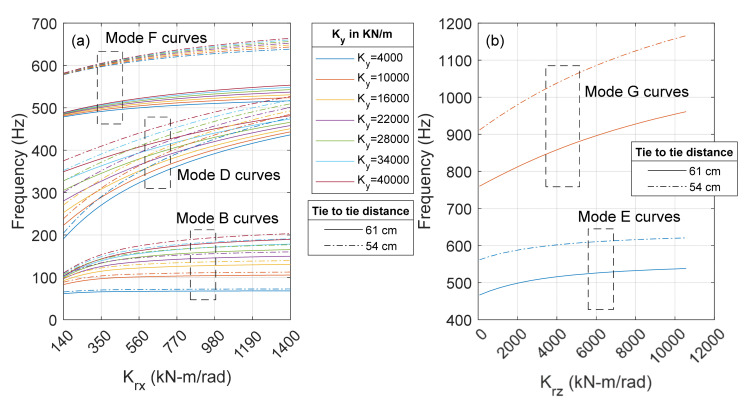
Sensitivity analysis. Effect of tie-to-tie distance on the resonant frequency of (**a**) modes B, D and F for different values of *K_y_* and *K_rx_*, and (**b**) modes E and G for a range of *K_rz_* coefficients.

**Figure 7 sensors-22-07447-f007:**
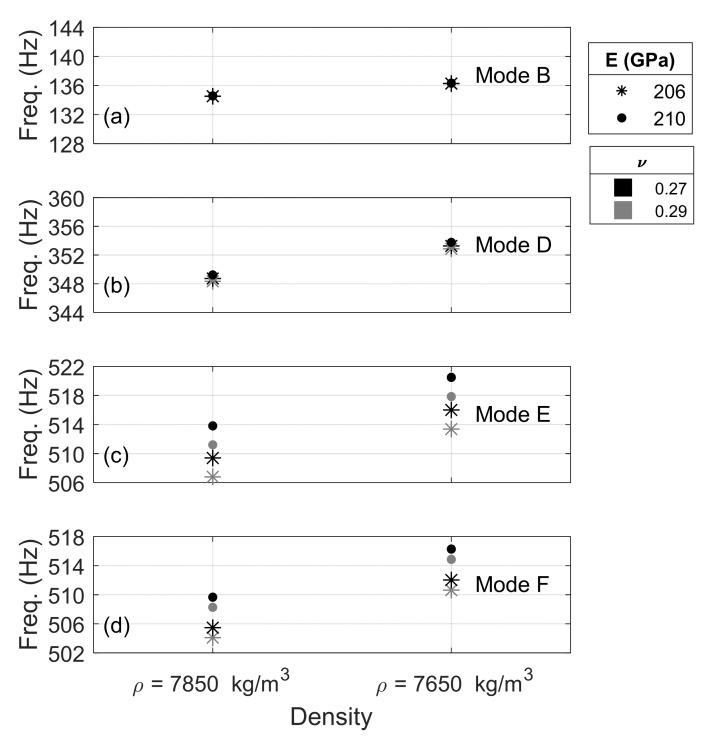
Sensitivity analysis. Effect of the rail material properties on the frequency of vibration of modes B, D, E, and F shown in subplots (**a**), (**b**), (**c**), and (**d**) respectively.

**Figure 8 sensors-22-07447-f008:**
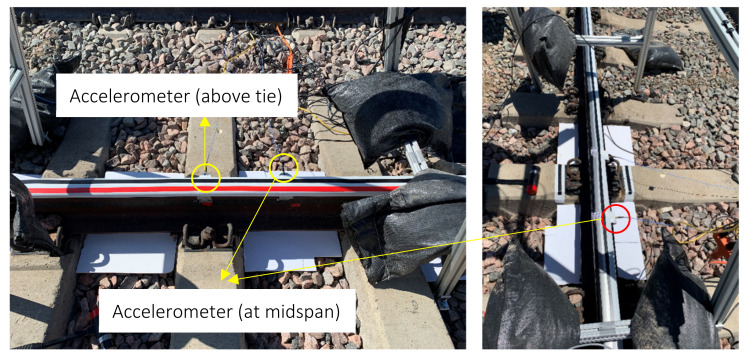
Close-up view of the rail on concrete ties tested in the field.

**Figure 9 sensors-22-07447-f009:**
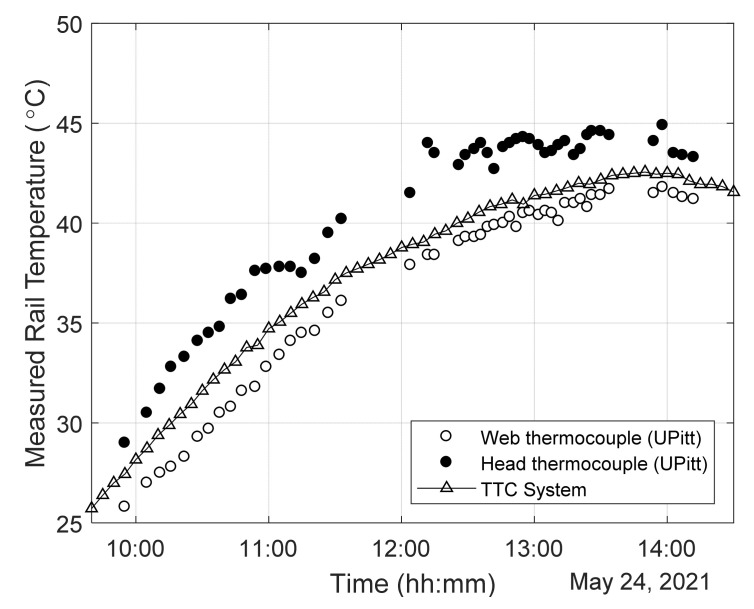
Temperatures of the track taken with three independent units. The lack of data between 11:30 AM and Noon was related to some setting adjustments not related with the content of this paper.

**Figure 10 sensors-22-07447-f010:**
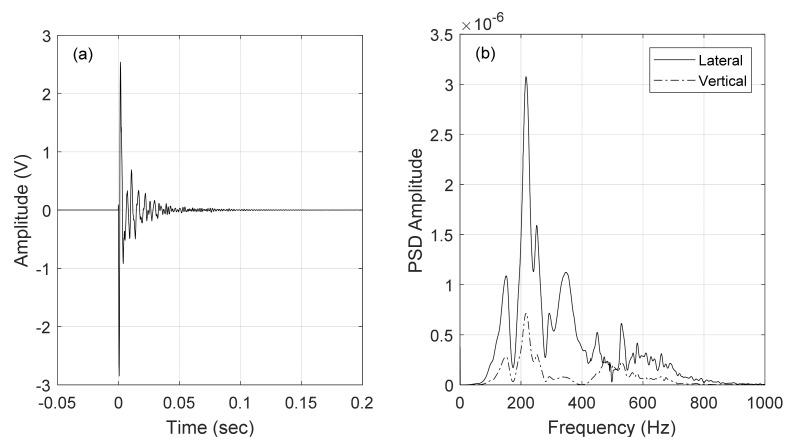
(**a**) Typical time-series associated with the lateral displacement and recorded by the accelerometers on the rail above the cross-tie. (**b**) Corresponding Power Spectral Density overlapped to the PSD of the vertical direction.

**Figure 11 sensors-22-07447-f011:**
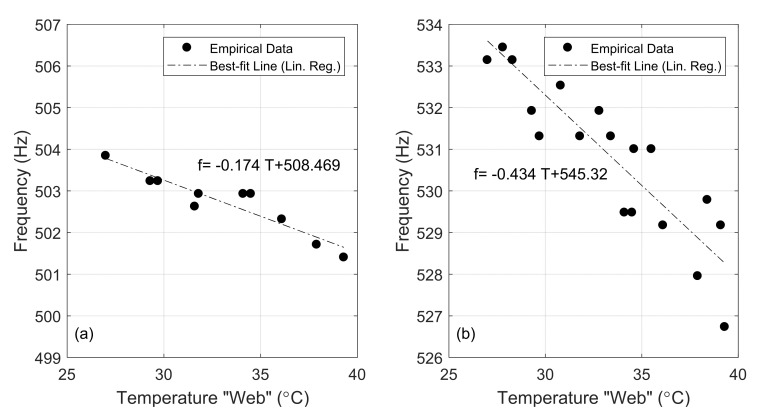
(**a**) Frequency peaks associated with (**a**)-peak around 500 Hz and (**b**)-peak around 530 Hz as a function of the web temperature recorded at the moment of the lateral impacts.

**Figure 12 sensors-22-07447-f012:**
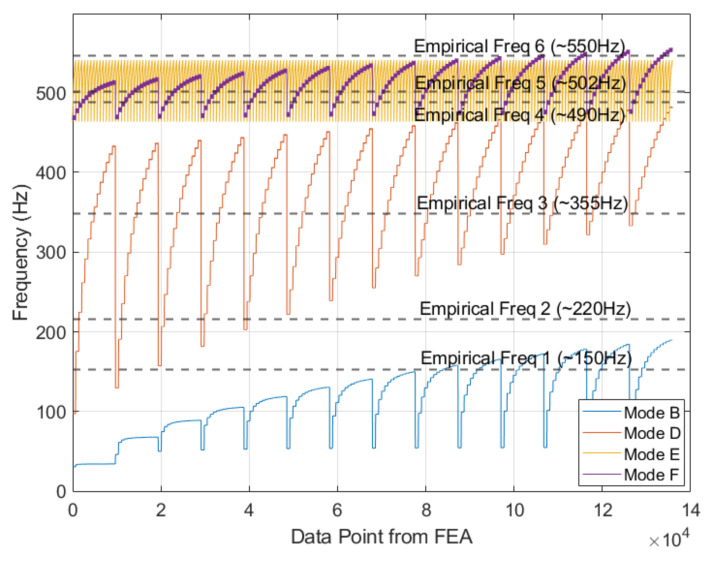
Day 1 Accelerometer Data vs. FEA. Note that 6 frequencies were extracted which are represented by the horizontal lines. The step-wise trend is caused by shift in boundary conditions. This is particularly noticeable on Mode D and parts of Mode B.

**Figure 13 sensors-22-07447-f013:**
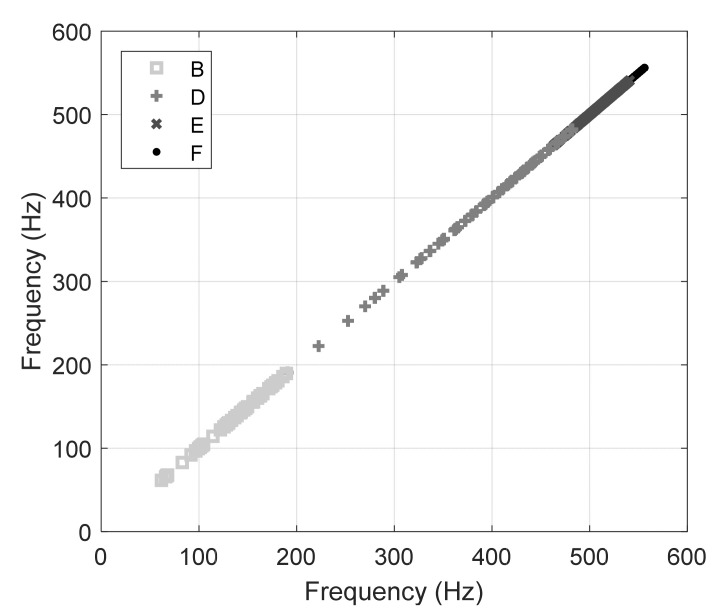
RFEA Mode Shape Distributions. Note the overlap between D, E, and F.

**Figure 14 sensors-22-07447-f014:**
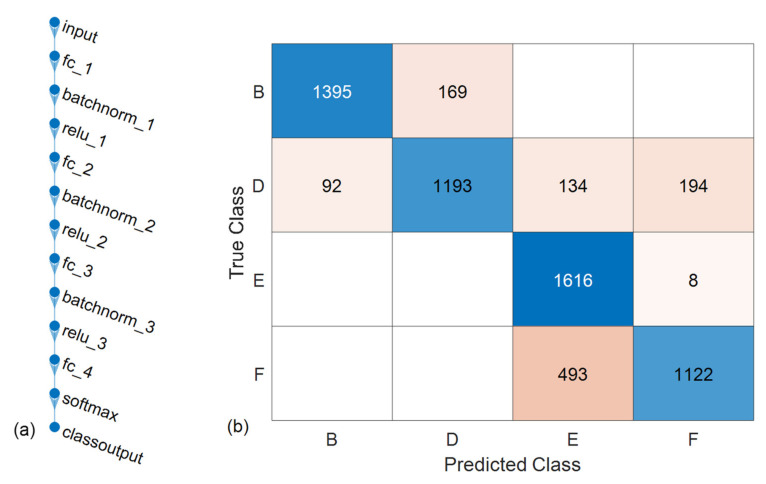
Mode Shape Classifier: (**a**) ANN Architecture; (**b**) Confusion Matrix.

**Figure 15 sensors-22-07447-f015:**
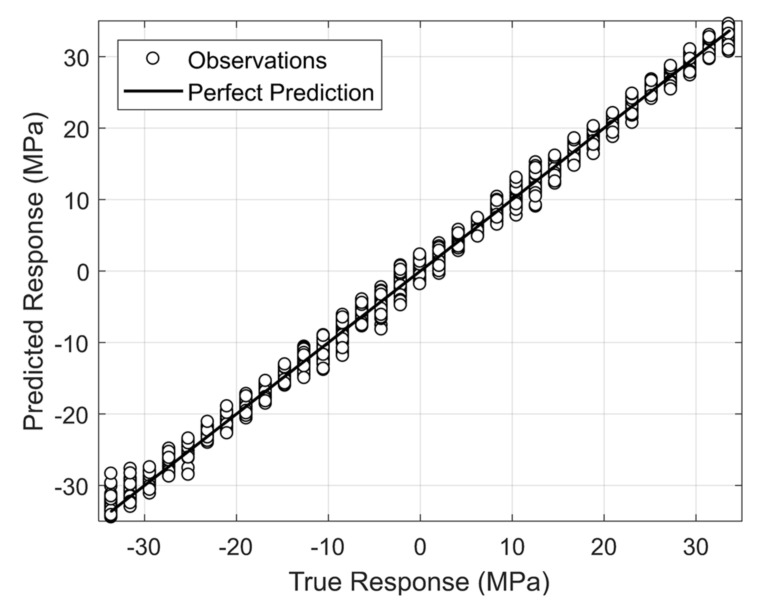
Testing the machine learning using data generated using finite element analysis. Actual stress vs. predicted stress.

**Figure 16 sensors-22-07447-f016:**
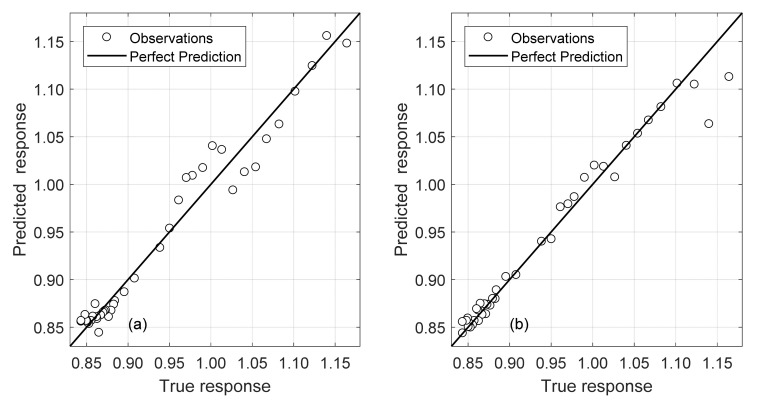
D1 Model Validation Predicted Vs Actual: (**a**) D1_SVM_F5_F6; (**b**) D1_GPR_F1_F6.

**Figure 17 sensors-22-07447-f017:**
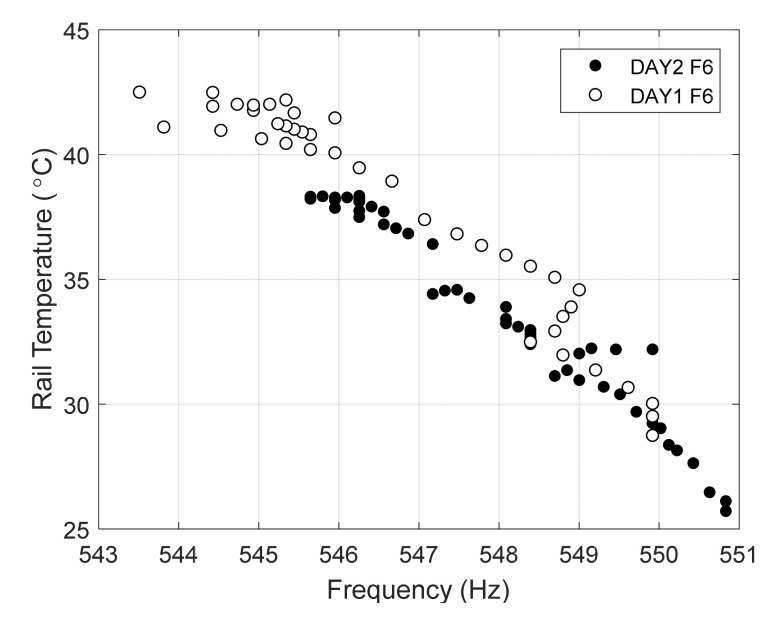
Day 1 vs. Day 2 Frequency 6.

**Figure 18 sensors-22-07447-f018:**
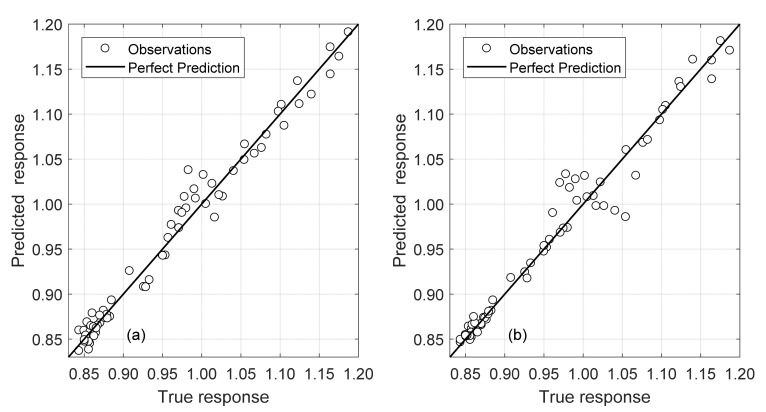
D12 Model Validation Predicted Vs Actual: (**a**) D12_ GPR_F1_F6; (**b**) D12_ GPR_F5_F6.

**Figure 19 sensors-22-07447-f019:**
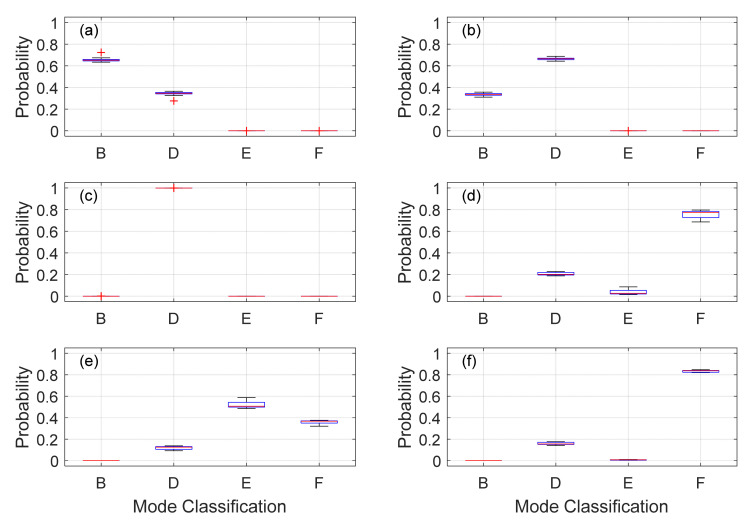
Mode-shape probability predictions of the accelerometer frequencies on Day 1 for peaks around (**a**)—150 Hz, (**b**)—220 Hz, (**c**)—350Hz, (**d**)—490 Hz, (**e**)—500 Hz, and (**f**)—550 Hz.

**Figure 20 sensors-22-07447-f020:**
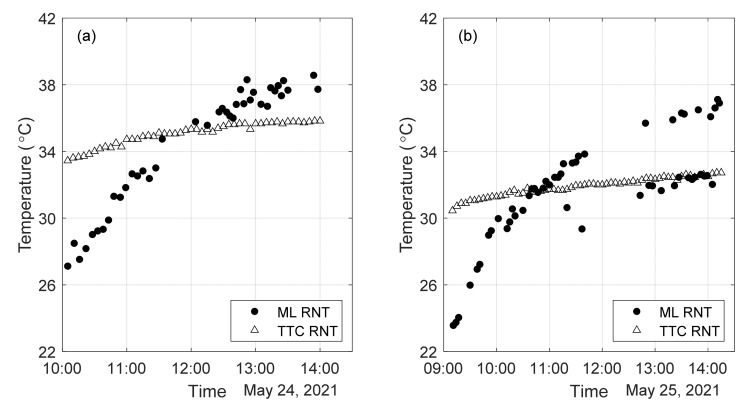
ML predictions based on the database obtained with the finite element analysis. The value of the predicted neutral temperature is overlapped to the independent estimation provided by TTC. (**a**) Results relative to Day 1; (**b**) Results relative to Day 2.

**Figure 21 sensors-22-07447-f021:**
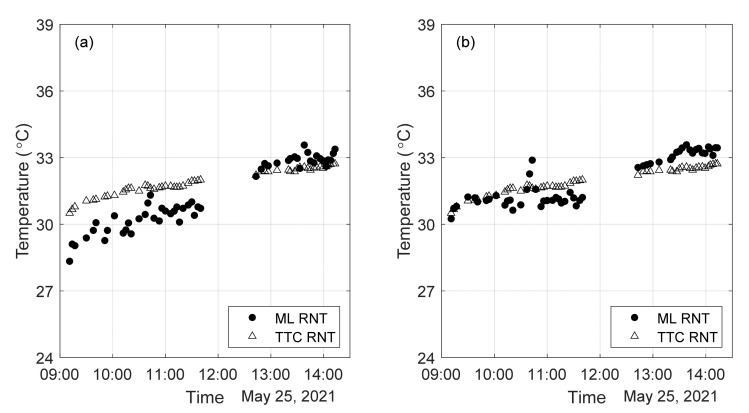
Testing the D1 on empirical data D2. RNT vs. Temperature for: (**a**) F1 through F6 Input; (**b**) F5 and F6 Input.

**Figure 22 sensors-22-07447-f022:**
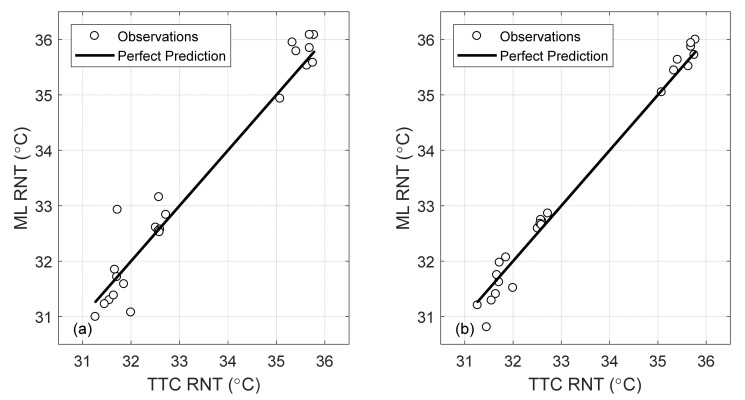
Testing the D12 on empirical data. Predicted RNT vs. Actual RNT for: (**a**) F1 through F6 Input; (**b**) F5 and F6 Input.

**Table 1 sensors-22-07447-t001:** Variables considered in the FE modeling of the rail for a rail on concrete ties and a rail on wood ties.

Track Type Based on Tie Material	Parameter	Minimum Value	Maximum Value	No. of Equal Increments for the Given Parameter
Concrete	*K_y_* (kN/m)	2000	40,000	14
	*K_rx_* (kN-m/rad)	70	1400	14
	*K_rz_* (kN-m/rad)	70	10,570	21
	Axial Stress (MPa)	−33.6	33.6	33
Wood	*K_y_* (kN/m)	1500	48,000	6
	*K_rx_* (kN-m/rad)	52.5	1680	6
	*K_rz_* (kN-m/rad)	11.2	2870	9
	Axial Stress (MPa)	−33.6	33.6	33

**Table 2 sensors-22-07447-t002:** Effect of the rail cross-section on the resonant frequencies at a given boundary conditions and at neutral.

	136 RE	141 RE
Mode B (Hz)	138.49	134.50
Mode D (Hz)	350.56	348.34
Mode E (Hz)	496.25	506.80
Mode F (Hz)	501.28	504.09

**Table 3 sensors-22-07447-t003:** Frequencies extracted from the two accelerometers on Days 1 and 2.

	F1	F2	F3	F4	F5	F6
Frequency	~150 Hz	~220 Hz	~350 Hz	~490 Hz	~500 Hz	~550 Hz

**Table 4 sensors-22-07447-t004:** Dataset summary.

Training/Validation Set	Test Set	Inputs (Hz)	Output	Description
RFEA	D12	Mode B, D, E, F	Stress (Mpa)	Numerical Finite-Element Dataset provided via ANSYS simulation. This is used to test on Days 1 and 2 (D12).
D1	D2	F5, F6 or F1-F6	*β* (No Unit)	This experimental dataset from Day 1 provides six frequencies of which the two highest or all six are used as inputs to determine a scaling factor, *β*, of the rail temperature to get RNT. Day 2 (D2) data is used as the test set.
D12	D12 *	F5, F6 or F1-F6	*β* (No Unit)	This experimental dataset combines Days 1 and 2 data (same rail location). Inputs and output are the same as the D1 Dataset. Test set is a random split from both days.

* Data used in D12 Test set are unseen from training/validation D12 set.

**Table 5 sensors-22-07447-t005:** RFEA Validation Results.

MODEL	*RMSE*	*R^2^*	*MSE*	*MAE*	ERROR (°C)
RFEA_Ensemble	0.654887	0.998927	0.428877	0.472164	0.19

**Table 6 sensors-22-07447-t006:** D1 Validation Results for F1-F6 and F5, F6 Model Inputs.

MODEL	*RMSE*	*R^2^*	*MSE*	*MAE*	ERROR (°C)
D1_GPR_F1_F6	0.016648	0.97	0.00027714	0.009082	0.27246
D1_SVM_F5_F6	0.012384	0.98	0.00015336	0.0070587	0.211761

**Table 7 sensors-22-07447-t007:** D12 Validation Results for F1–F6 and F5, F6 Model Inputs.

MODEL	*RMSE*	*R^2^*	*MSE*	*MAE*	ERROR (°C)
D12_GPR_F1_F6	0.016101	0.972	0.00026095	0.011646	0.355
D12_GPR_F5_F6	0.016640	0.972	0.00027934	0.009949	0.298

**Table 8 sensors-22-07447-t008:** D1 performance metrics on empirical data.

MODEL	*RMSE*	*R^2^*	*MSE*	*MAE*	ERROR (°C)
D1_GPR_F1_F6	0.03641	0.86	0.001326	0.02978	0.893
D1_SVM_F5_F6	0.02303	0.94	0.0005304	0.01826	0.548

**Table 9 sensors-22-07447-t009:** D12 mean performance metrics on empirical data across five splits.

MODEL	*RMSE*	*R^2^*	*MSE*	*MAE*	ERROR (°C)
D12_GPR_F1_F6	0.0095174	0.986	9.20152 × 10^−5^	0.00728306	0.218
D12_GPR_F5_F6	0.00741366	0.99	5.50406 × 10^−5^	0.00548996	0.165

## Data Availability

Some or all data, models, or code that support the findings of this study are available from the corresponding author upon reasonable request.

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
