# Peer review of "Vibration-Based Approach to Measure Rail Stress: Modeling and First Field Test"

_sensors, 2022, doi:10.3390/s22197447_

Round 1
Reviewer 1 Report
The present study discusses a non-invasive and non-destructive technique to estimate stresses in continuous welded rails by adopting a finite element model on which the authors performed detailed sensitivity analyses. From this model, vibration data are retrieved to train and compare several supervised data-driven machine learning models based on modal frequencies extracted via the power spectral density method. These models were subsequently validated on real vibration acceleration data of a railroad section tested in Colorado, arguing about the feasibility of the proposed approach. The present study is well-written and organized, the results sound interesting, and I recommend the work to be accepted for publication in your journal if and only if the authors introduce the following minor revisions and changes:
• Within the introduction, when nondestructive evaluation techniques are presented, and in order to improve and complete the state-of-art literature review on the use of artificial intelligence methods in civil engineering, the authors are suggested to take into consideration the present work as well:
Marasco, G., Rosso, M. M., Aiello, S., Aloisio, A., Cirrincione, G., Chiaia, B., & Marano, G. C. (2022). Ground Penetrating Radar Fourier Pre-processing for Deep Learning Tunnel Defects’ Automated Classification. In International Conference on Engineering Applications of Neural Networks (pp. 165-176). Springer, Cham.
Aloisio, A., Rosso, M. M., & Alaggio, R. (2022). Experimental and Analytical Investigation into the Effect of Ballasted Track on the Dynamic Response of Railway Bridges under Moving Loads. Journal of Bridge Engineering, 27(10), 04022085.
• In the abstract at line 21 it is reported that “Three ML models were developed […]”, however in section 5 at lines 575-577 it was written that “Algorithms used were Linear Regression, Random Forest, LightGBM, XGBoost, CatBoost, ANN, and Ensembling”. Does the “Three” in the abstract refer to the three datasets mentioned in line 550? Please revise that sentence to avoid misunderstanding.
• For the sake of completeness, please already mention in the introduction which kind of machine learning models have you implemented. Otherwise, it is not clear for the reader during the reading initial parts, because it is always referred generally speaking to machine learning models without giving a clear idea of which techniques have been adopted in the current study until reaching the final sections.
Author Response
Reviewer 1
Comments to the Author
The present study discusses a non-invasive and non-destructive technique to estimate stresses in continuous welded rails by adopting a finite element model on which the authors performed detailed sensitivity analyses. From this model, vibration data are retrieved to train and compare several supervised data-driven machine learning models based on modal frequencies extracted via the power spectral density method. These models were subsequently validated on real vibration acceleration data of a railroad section tested in Colorado, arguing about the feasibility of the proposed approach. The present study is well-written and organized, the results sound interesting, and I recommend the work to be accepted for publication in your journal if and only if the authors introduce the following minor revisions and changes:
AUTHORS’ response.
We deeply appreciate the time and effort you have spent in our manuscript entitled “Artificial Intelligence and Vibration Data to Determine Stress in Rails: Modeling and Field Tests”. And thanks very much for your careful review and your constructive comments. These comments are valuable and helpful for revising and improving our paper. Based on these comments, we have tried our best to make careful revisions on the original manuscript which we hope can meet with your approval. This rebuttal letter contains our replies to your comments. To ease your work, we used black bold font for the referee’s comments, red font for our replies, and bold red italics font for text excerpted from the manuscript. We used MS Word Track Change tool to revise our original manuscript. For convenience a copy of the revised manuscript with and without Track Changes is provided.
- Within the introduction, when nondestructive evaluation techniques are presented, and in order to improve and complete the state-of-art literature review on the use of artificial intelligence methods in civil engineering, the authors are suggested to take into consideration the present work as well:
Marasco, G., Rosso, M. M., Aiello, S., Aloisio, A., Cirrincione, G., Chiaia, B., & Marano, G. C. (2022). Ground Penetrating Radar Fourier Pre-processing for Deep Learning Tunnel Defects’ Automated Classification. In International Conference on Engineering Applications of Neural Networks (pp. 165-176). Springer, Cham.
Aloisio, A., Rosso, M. M., & Alaggio, R. (2022). Experimental and Analytical Investigation into the Effect of Ballasted Track on the Dynamic Response of Railway Bridges under Moving Loads. Journal of Bridge Engineering, 27(10), 04022085..
AUTHORS’ response.
Thanks for introducing these references. These two are now added in the introduction.
- In the abstract at line 21 it is reported that “Three ML models were developed […]”, however in section 5 at lines 575-577 it was written that “Algorithms used were Linear Regression, Random Forest, LightGBM, XGBoost, CatBoost, ANN, and Ensembling”. Does the “Three” in the abstract refer to the three datasets mentioned in line 550? Please revise that sentence to avoid misunderstanding.
AUTHORS’ response. That is correct – The three best models of those ML algorithms experimented with (in section 5) were chosen for our three datasets. This was revised on line 21 to the following: “Three datasets were prepared and fed to ML models developed using hyperparameter search optimization techniques and k-fold cross validation to infer the stress or the RNT.”
- For the sake of completeness, please already mention in the introduction which kind of machine learning models have you implemented. Otherwise, it is not clear for the reader during the reading initial parts, because it is always referred generally speaking to machine learning models without giving a clear idea of which techniques have been adopted in the current study until reaching the final sections.
AUTHORS’ response. The following has been added on page 3/29 in the Introduction after mention of our previous work and improvements we have made on this work: “These MLAs consisted of Linear Regression, Extreme Gradient Boosting (XGBoost), CatBoost, Ensembling, Artificial Neural Networks (ANN), Support Vector Machines (SVM), Decision Trees, and Gaussian Process Regression (GPR).”

Reviewer 2 Report
The authors presented a vibration-based method coupled to ML and FEM analysis for the non-invasive neutral temperature inspection of the continuous rails. The proposed strategy was tested both numerically and experimentally on field data. This work seems comprehensive and valuable for the actual application. The manuscript is content-rich, well-organized, and well-written. Therefore, this manuscript is suggested to have a minor revision before being accepted in my view.
The detailed comments can be seen below:
1. All the figures and tables are suggested to be centered uniformly, such as Fig 1, 2, 3, table 2, etc.
2. “RE 136, RE 141” mentioned in this manuscript seems to lack of explanation of the meaning.
3. Line 361, the authors mentioned, “Both accelerometers were triggered via an instrumental hammer and sampled at 10 kHz.” Can the operator keep the amplitude of the hammer consistent during each testing, where is the impact position, and what was the lasting time for each test?
4. In figure 11, labels (a)(b) are suggested to keep in the upper left corner of this figure.
5. Line 561, this sentence should be left aligned.
Author Response
Reviewer 2
Comments to the Author
The authors presented a vibration-based method coupled to ML and FEM analysis for the non-invasive neutral temperature inspection of the continuous rails. The proposed strategy was tested both numerically and experimentally on field data. This work seems comprehensive and valuable for the actual application. The manuscript is content-rich, well-organized, and well-written. Therefore, this manuscript is suggested to have a minor revision before being accepted in my view.
AUTHORS’ response.
We deeply appreciate the time and effort you have spent in our manuscript entitled “Artificial Intelligence and Vibration Data to Determine Stress in Rails: Modeling and Field Tests”. And thanks very much for your careful review and your constructive comments. These comments are valuable and helpful for revising and improving our paper. Based on these comments, we have tried our best to make careful revisions on the original manuscript which we hope can meet with your approval. This rebuttal letter contains our replies to your comments. To ease your work, we used black bold font for the referee’s comments, red font for our replies, and bold red italics font for text excerpted from the manuscript. We used MS Word Track Change tool to revise our original manuscript. For convenience a copy of the revised manuscript with and without Track Changes is provided.
- All the figures and tables are suggested to be centered uniformly, such as Fig 1, 2, 3, table 2, etc.
AUTHORS’ response.
Noted.
- “RE 136, RE 141” mentioned in this manuscript seems to lack of explanation of the meaning.
AUTHORS’ response.
Great comment. The definition of RE 136 is provided in the first paragraph of section 2.2.1. As mentioned there, the value 136 corresponds to the weight of the section per unit length, thus the authors didn’t repeat that for the RE 141 type.
- Line 361, the authors mentioned, “Both accelerometers were triggered via an instrumental hammer and sampled at 10 kHz.” Can the operator keep the amplitude of the hammer consistent during each testing, where is the impact position, and what was the lasting time for each test?
AUTHORS’ response.
As the impact were applied manually, there is no guarantee that the amplitude of the impacts was consistent from one hit to another. Therefore, there might be some variations in the impulse associated with each impact. To address other questions regarding the impact position and the protocol to apply hits, the following is added in section 3.1. below figure 8. “Impacts were applied on the field side of the rail just behind the midspan and tie location where the sensors were bonded. Only one of the two spots was hit during a single measurement.”
In addition, as can be inferred from the setup nature, each measurement lasted for a fraction of a second. Duration of each collected signal is shown in figure 10.a.
- In figure 11, labels (a)(b) are suggested to keep in the upper left corner of this figure.
AUTHORS’ response.
Thanks for the comment. However, to be consistent with the other provided figures (Fig. 14, 16, 18, 20, 21, and 22), authors prefer to keep it as is.
- Line 561, this sentence should be left aligned.
AUTHORS’ response.
Noted.
